# Tidal Sediment Supply Maintains Marsh Accretion on the Yangtze Delta despite Rising Sea Levels and Falling Fluvial Sediment Input

**Peng Li [1], Benwei Shi [2,3], Guoxiang Wu [3,4], Wenxiang Zhang [3,*], Sijian Wang [1], Long Li [1], Linghao Kong [5] and Jin Hu [3]**

[1] East China Sea Forecasting and Hazard Mitigation Center, Ministry of Natural Resources, Shanghai 200081,China

[2] Key Laboratory of Coastal Science and Integrated Management, Ministry of Natural Resources, Qingdao 266061, China

[3] State Key Lab of Estuarine and Coastal Research, East China Normal University, Shanghai 200241, China

[4] College of Engineering, Ocean University of China, Qingdao 266100, China

[5] Yantai Coastal Zones Geological Survey, China Geological Survey, Yantai 264000, China

[*] Correspondence: wxzhang@sklec.ecnu.edu.cn

**Abstract:** Tidal marshes are among the world's most valuable ecosystems; however, they are increasingly threatened by rising sea levels and a decline in fluvial sediment supply. The survival of a tidal marsh under these twin threats depends upon the net input of tidal sediments, because this will determine the deposition rate. The rate of relative sea level rise currently affecting the Yangtze Delta is rapid (~4 mm/a), and the sediment discharge from the Yangtze River has decreased by >70% over recent decades. In order to improve our understanding of the response of the marshes in the turbid zone of the Yangtze Estuary to these changing environmental conditions, we measured sediment transport in and out of a tidal basin and calculated the deposition rate over eight tidal cycles covering different tidal ranges during the summer and winter seasons. The suspended sediment concentration (SSC) during the flood phase of the tidal cycle (average = 0.395 kg/m$^3$) was markedly higher than that during the ebb (average = 0.164 kg/m$^3$), although water transport during the flood tide was almost equivalent to that during the ebb. As a result, ~40% of the sediment inflow during the flood phase was retained within the marsh. This reason is mainly attributable to the dense marsh vegetation, which attenuates waves and currents and to which the sediments adhere. The annual deposition rate in the marsh was approximately 6.7 mm/a. These findings indicate that under the combined influence of sea level rise and river sediment supply reduction, the sediment transport through the turbidity maximum zone of the Yangtze River estuary could maintain the relative stability of the marsh area to a certain extent.

**Keywords:** sediment transport; tidal marsh; sea level rise; marsh survival; tidal creek; Yangtze Estuary



## 1. Introduction

Tidal marshes are unique and highly productive ecosystems [1]. Not only do they provide irreplaceable habitats for aquatic and terrestrial animals and migratory birds, but they also protect shorelines from storms, filter pollutants, reduce flood risks, replenish groundwater, and maintain biodiversity [2–4]. Unfortunately, tidal marshes—and in particular deltaic marshes—are threatened by the accelerating rise in sea levels and a decline in fluvial sediment supply [5]. The rate of global sea level rise increased from 1.4 mm/a over the period 1880–1993 to 2.9 mm/a over the period 1993–2010 [6] and will continue to accelerate to reach 3–15 mm/a by the end of this century [7]. In addition, sediment discharge from the world's rivers has decreased substantially, owing mainly to dam construction over recent decades [8]. The survival of tidal marshes will majorly depend upon

the residual sediment transport into the marshes, because this determines the deposition rate. Previous studies have shown that the tidal creek in salt marsh is the main channel for the exchanges of water, sediment and nutrients between the tidal basin and the subtidal zone [9–11], which controls the development of the geomorphology and ecological in salt marsh partially [12,13]. At the present, geomorphological evolution of the tidal creek is influenced increasingly by human activities and sea level rise [14–16].

Previous studies have evaluated the survival potential of marshes by comparing the rates of marsh deposition and sea level rise [17] or calculating a critical suspended sediment concentration (SSC) [18]. For example, numerical models have predicted that a typical tidal marsh would be inundated by the end of this century if the SSC was <0.02 kg/m$^3$ [19–21]. However, little is known about sediment transport through tidal creeks and its relationship to marsh survival under a regime of rising sea level and falling fluvial sediment supply. Thus, it is necessary for an integrated investigation of hydrodynamic and sediment transport in the tidal creek to understand the reason of the degradation of deltaic salt marsh.

The rate of relative sea level rise on the Yangtze Delta is currently ~4 mm/a, and sediment discharge from the Yangtze to the sea has decreased by 70% over recent decades [5]. The Jiuduansha Shoal is a tidal wetland island within the turbidity maximum zone of the Yangtze Estuary [22]. Tidal creek systems are well developed on the Jiuduansha Shoal (Figures 1 and 2–4 in [23]). The tidal marsh is composed mainly of *Spartina alterniflora* and *Scirpus mariqueter*, which capture the suspended sediments and slow the flow of water across the marsh, increasing deposition as a result [24,25]. The Jiuduansha Shoal is located at the front of the Yangtze Estuary and is an ideal site to study whether tidal marshes can survive through the deposition of sediment transported via tidal creeks under the conditions of rapid sea level rise and a decline in fluvial sediment supply.

In this study, we aim to estimate the accretion rate on the Jiuduansha tidal marsh and compare this with the rate of sea level rise. Our objectives are to (1) measure flow velocity and SSC at the mouth of a tidal creek system over neap and spring tidal cycles in different seasons, (2) calculate net sediment transport into the tidal marsh over each tidal cycle, (3) delineate the boundary of the tidal basin, and (4) estimate the annual deposition rate on the tidal marsh. This study is helpful to understand hydrodynamic mechanics of morphological evolution in the Jiuduansha Shoal in the Yangtze Estuary under the influence of the sharp decrease of Yangtze River sediment discharge into the sea. Our study can provide support for wetland resource management and the scientific decision making of National Wetland Protection Regulations in the conservation of China's wetlands.

## 2. Field Setting

The study area was located on the Jiuduansha Shoal, which is the largest uninhabited alluvial island in the Yangtze Estuary, China, and consists of the Upper, Middle, and Lower shoals (Figure 1C). The Jiuduansha Shoal is located within the turbidity maximum zone of the Yangtze Estuary [14]. Since its formation in the 1950s, the Jiuduansha Shoal has been expanding gradually [23,26]. In 1997, an area of 0.9 km$^2$ of Phragmites australis and *Spartina alterniflora* were planted on the Middle and Lower shoals [27,28]. From 1958 to 2005, the area and volume of the Jiuduansha Shoal above the 5 m isobath increased by 96% and 156%, respectively, the intertidal area and volume increased by 331% and 504%, respectively, the maximum elevation above the lowest astronomic tide increased from 0.3 to 4.9 m, and the proportion of the total intertidal area covered by salt marsh increased from 0% to 40% [23]. In 2005, the area above the 5 m isobath was 413 km$^2$, and the area above the 0 m isobath was 168 km$^2$ [23]. In 2020, the area of the Jiuduansha Shoal exposed at high and low tide was $100 \pm 4$ and $155 \pm 6$ km$^2$, respectively [29]. The Jiuduansha Shoal has an irregular semidiurnal tide, and the average tidal range is 2.67 m, the maximum tidal range is 4.42 m, the annual average SSC in the surface layer is 0.54 kg/m$^3$, and the average salinity is 3.89 [30,31]. During the field measurement period, the mean and median grain sizes of eight suspended sediment samples from adjacent waters were $10.3 \pm 0.7$ µm

(ranging from 9.6 to 11.6 μm) and 6.3 ± 0.5 μm (ranging from 5.6 to 7.0 μm), respectively. The mean and median grain sizes of 22 surface sediment samples collected in the Lower Shoal tidal marsh during the observation period were 25.6 ± 8.8 μm (ranging from 13.4 to 44.8 μm) and 19.0 ± 8.4 μm (ranging from 8.6 to 36.9 μm), respectively. The median grain size of the Jiuduansha Shoal was equivalent to the median size of the sediment output from the Yangtze River. The maximum current velocity in the North Passage and South Passage near the Jiuduansha Shoal was 2–2.5 m/s [32]. The sediment distribution on the Jiuduansha Shoal was ring-shaped. The sediment in the central tidal marsh zone is composed mainly of fine, clayey silt, and the sediment coarsens gradually outward and transitions to fine sand on the outer bare flat [33].

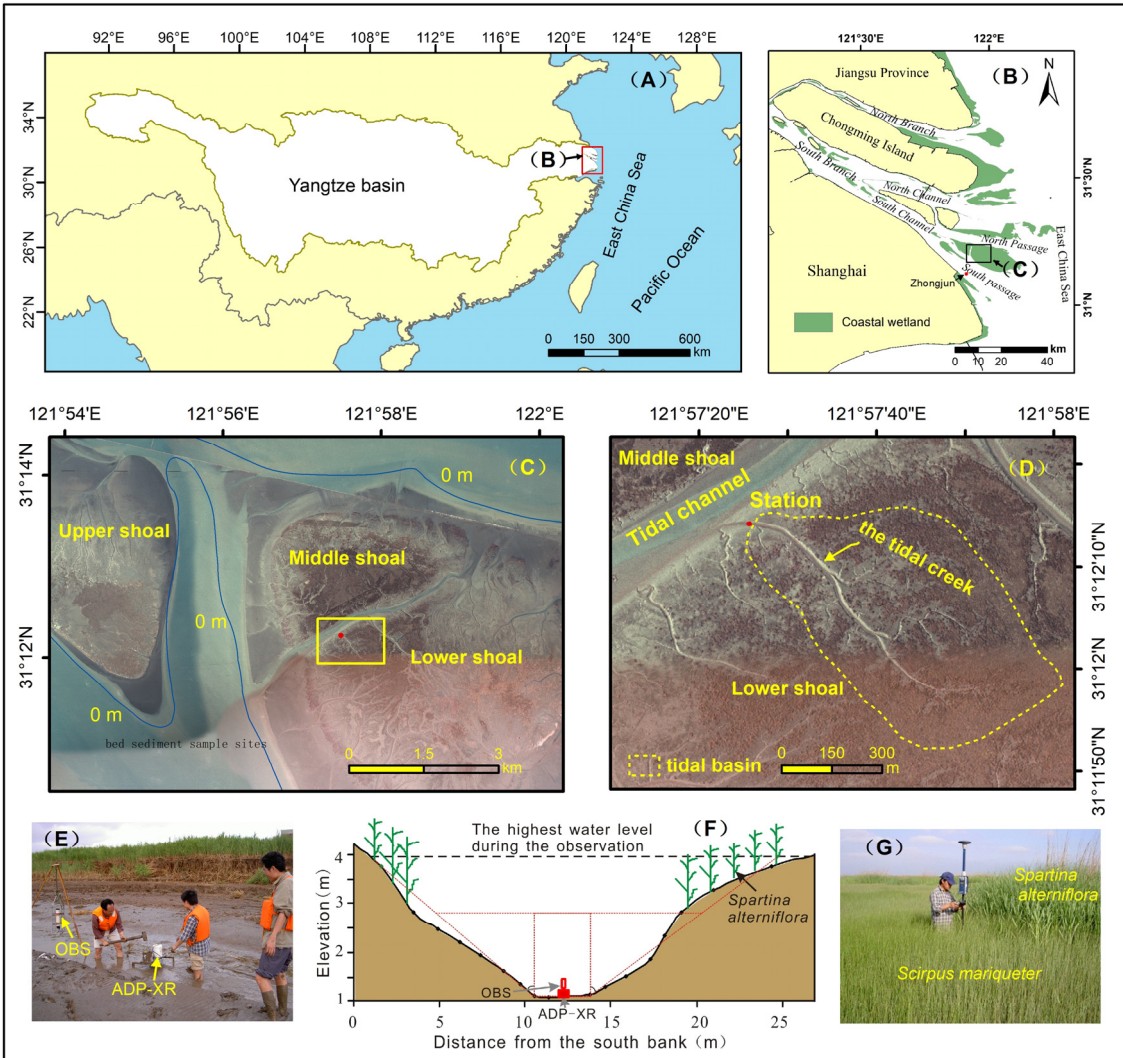

**Figure 1.** (**A**) Map of the study area showing the location of (**B**) the Jiuduansha Shoal in the Yangtze Estuary. (**C**) High-resolution satellite images showing the tidal creek (Upper, Middle, and Lower shoals), and (**D**) details of the Lower Shoal, showing the locations of data measurements. (**E**) Photograph showing field measurements being collected in the creek in summer. (**F**) Instrument layout in the middle of the tidal creek. (**G**) Photograph showing field observations being conducted on the marsh.

The field measurements in the tidal creek were made on the Lower Shoal, and the distance from the observation station to the mouth of the creek was ~100 m. The bottom width of the creek cross section was 3.2 m, and its upper width was 26.9 m (Figure 1F). The creek ran northwest–southeast, and the length of the main channel was ~1300 m (Figure 1).

The tidal basin area was ~0.42 km$^2$ (Figure 1D). The tidal marsh in the study area was covered mainly by *Spartina alterniflora* and *Scirpus mariqueter*; the former grows to a height of ~1 m and covers ~95% of the marsh, whereas the latter grows to a height of 0.4 m and covers ~50% of the area, respectively.

## 3. Materials and Methods

### 3.1. Field Measurement

Field measurements of current velocity, water depth, and turbidity were carried out over a spring tide between 25 and 28 June 2006 (in summer), and over spring and neap tides between 29 December 2006, and 4 January 2007 (in winter), at a fixed site in the tidal creek on the Lower Shoal. We measured current velocity and direction, as well as echo intensity, at five levels (0.4, 0.6, 0.8, and 1.0 m above the seabed, and at the water surface ≥1.2 m above the seabed), using an ADP-XR (acoustic Doppler current meter, Argonaut-XR, 3.0 MHz, A YSI Environmental Company, San Diego, CA, USA). Turbidity and salinity were measured 0.2 m above the seabed using an OBS-3A (optical backscatter sensor, CompbellScientific Inc., Logan, UT, USA) in summer and at 0.2 and 0.4 m above the seabed using two OBS-3As in winter. The pressure-based water levels were determined using both the ADP-XR and OBS-3A sensors relative to the sediment surface as a reference height. The OBS-3A measurements were taken at 1-min sampling intervals. The ADP-XR measurements were taken at 5-min sampling intervals in summer and at 2-min sampling intervals in winter. Both the 5- and 2-min measurements were burst-averaged, the burst period being 1 s. In order to establish the regression relationship between the OBS-3A readings and the SSC, we collected in situ water samples to use for OBS-3A calibration in the laboratory [34].

The elevations were surveyed across the tidal creek and on the tidal marshes using an RTK-GPS (real-time kinematic global positioning system; Figure 1G). The water surface was always lower than the banks of the tidal creek at our fixed monitoring site, i.e., the flood tides did not overflow the banks of the creek. The water and sediment data from the Yangtze River (Datong Station) were obtained from the Yangtze River Water Resources Committee. The wind speed and direction data from Dajishan Station and the tidal range and height data from Zhongjun Station were obtained from the Forecast Center for the East China Sea, State Oceanic Administration of China.

Due to the limitation of observation cost, only 8 tidal cycles were observed, including spring and neap tides in winter and summer. In general, the observation results can represent the hydrodynamic characteristics and siltation characteristics of the tidal flat in winter and summer. The spring and fall characteristics were between winter and summer. In this paper, we focused on the developmental and evolutionary trend of tidal marshes. In the absence of long-term and continuous observation data, the influence of tidal cycle and seasonal errors would be filtered out in the calculation of one-year time. Therefore, the annual sedimentation rate estimated from these 8 tidal cycles is relatively scientific and credible.

### 3.2. Calibration of SSC and Sediment Transportation

The turbidity signals were converted into SSC values via laboratory calibration using the in situ water samples. The regression between SSC and turbidity was established and yielded a correlation coefficient greater than 0.99 (Figure 2). The linear equations for both summer and winter are as follows (Figure 2):

$$C = 0.0014\,T - 0.0192,\ r = 0.998,\ n = 21,\ p < 0.001\ \textbf{(in summer)} \tag{1}$$

$$C = 0.0014\,T - 0.0247,\ r = 0.998,\ n = 21,\ p < 0.001\ \textbf{(in winter)} \tag{2}$$

where C is the calibrated SSC value, T are the OBS turbidity value, r is the regression coefficient, *n* is the number of observations, and *p* is the level of significance.

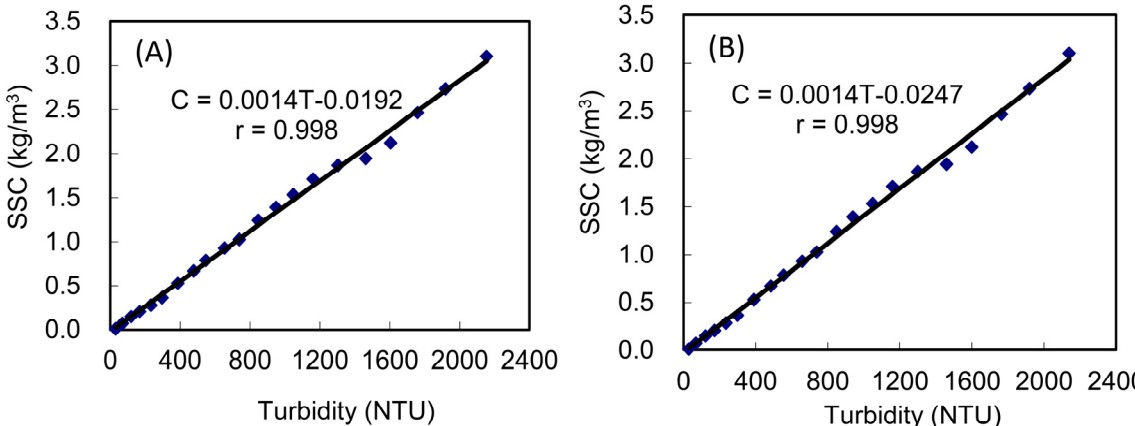

**Figure 2.** (**A**) Relationship between SSC (C) and OBS turbidity (T) in summer, and (**B**) Relationship between SSC (C) and OBS turbidity (T) in winter.

The echo intensity recorded by the ADP-XR is proportional to the concentration of suspended sediment and can be used to estimate SSC levels [35]. Therefore, we converted the ADP-XR echo intensity data into SSC values based on the comparison between the ADP-XR echo intensities and OBS-3A turbidities from the same intervals [34,35], and the relationship between OBS-3A turbidity and SSC (Figure 2). The relationship between the echo intensity from the ADP-XR and the SSC were established for both summer and winter as follows:

$$C = 0.00013\ e^{0.0979E1}, r = 0.90, n = 351, p < 0.001\ \textbf{(in summer)} \tag{3}$$

$$C = 0.00031\ e^{0.1041E2}, r = 0.87, n = 879, p < 0.001\ \textbf{(in winter)} \tag{4}$$

where C is the SSC value ($kg/m^3$), E is the echo intensity (dB), r is the regression coefficient, $n$ is the number of observations, and $p$ is the level of significance. Using these two equations, we calculated the SSC at the five heights above the seabed.

To simplify the calculation of the discharge of water and suspended sediment through the tidal creek, the cross section of the creek was regarded as a regular trapezoid (Figure 1F). When the water depth was <0.2 m, the SSC and current velocity could not be observed during either the flood or ebb tide periods. However, as this time was relatively short, the suspended sediment flux was relatively small when compared with the complete tidal cycle. When the water depth was >0.2 m, the SSC and current velocity in the blind cell were calculated the same as 0.2 m layer.

Owing to the limited number of instruments, observations were carried out only in the middle of the tidal creek and not on the slopes at either side. Assuming that the current velocity was constant at the same level across the tidal creek, in theory, the slopes on both sides of the bank would have a friction effect on the current, thus reducing the current velocity near the slope and affecting the SSC. However, according to our on-site observations, this impact was limited; only a few tens of centimeters were affected, whereas the average width of the tidal creek was 15 m. In other words, the calculation based on the above assumptions may have generated a value that is slightly larger than the actual cross-sectional average current velocity, but the difference between the two was probably insignificant. Furthermore, the aim of this study was to compare the discharge of water and sediment between the flood and ebb periods, spring and neap tides, and summer and winter. This paper focused on the developmental and evolutionary trends of tidal flat. Although there are errors in the calculation of the discharge sediment of each tidal cycle during the observation period, the effects of these errors will be filtered out in the calculation of one-year time, thus, the error of each tidal cycle is not estimated in this paper. In short, in the absence of more detailed data, the above assumptions were practical and appropriate and had little impact on the aims of our study.

The average current velocity (ua) of the tidal creek was calculated using the ADP-XR measurements of current velocity and water depth (see Section 3.1). The cross-sectional area (A) of the tidal creek was calculated as follows:

$$A = 3.2\,h + 0.5\,h^2 \times \tan 74° + 0.5\,h^2 \times \tan 76°$$ (5)

where 3.2 is the width at the bottom of the tidal creek, h is the instantaneous water depth, 74° is the angle of the south bank of the tidal creek, and 76° is the angle of the north bank. The cross section of the creek varied with the water depth; therefore, to simplify the calculation of the cross-sectional area, it was divided into five trapezoids (see Section 3.1, Figure 1F).

The water discharge ($Q_w$) through the tidal creek to the tidal marsh was calculated as follows:

$$Q_w = \int_0^t u_a A \, \mathrm{d}t$$ (6)

where $A$ is the cross-sectional area of the tidal creek and t is time. The SSC was divided into five layers according to water depth, each representing the SSC at these different water depths, and the suspended sediment discharge ($Q_s$) transported through the tidal creek was calculated as follows:

$$Q_s = \int_0^t u_a A C \, \mathrm{d}t$$ (7)

where $C$ is the SSC.

### 3.3. Tidal Basin Area

First, we delineated the boundary of the tidal basin using surveyed elevation data and the distribution pattern of tidal creek systems shown in the high-resolution satellite images (Figure 1C). Next, we calculated the area of the tidal basin using the ArcGIS 10.4 software package developed by the Environmental Systems Research Institute. The area of the tidal basin was ~0.42 km$^2$ (Figure 1D).

## 4. Results

### 4.1. Tidal Current

The current directions during the flood and ebb tide periods were 83° ± 5° and 258° ± 5°, respectively (Figure 2B), consistent with the orientation of the tidal creek. In addition, the current direction did not change with depth. The ebb tide duration (266 min) was significantly longer than that of the flood tide (200 min; Table 1). The horizontal current velocity varied from 0 to 84 cm/s, but the vertical variations in horizontal current velocity were relatively small (<10 cm/s; Figure 3). The greatest differences in the tidal current velocity occurred at the beginning of the flood tides and at the end of the ebb tides.

**Table 1.** Duration, time–depth-averaged flow velocity, and average SSC during flood and ebb phases.

| Season | Tidal Cycle | Maximum Water Depth (m) | Duration (min) | | Flow Velocity (cm/s) | | Average SSC (kg/m³) | | Net Deposition (kg) | Net Sediment Discharge Rates (%) |
|---|---|---|---|---|---|---|---|---|---|---|
| | | | Flood | Ebb | Flood | Ebb | Flood | Ebb | | |
| Summer | Tide 1 | 1.99 | 170 | 255 | 11.7 | 8.9 | 0.365 | 0.034 | 4606 | 85.5 |
| | Tide 2 | 2.91 | 255 | 282 | 32.9 | 57.5 | 0.377 | 0.342 | 15,002 | 30.3 |
| | Tide 3 | 1.92 | 145 | 270 | 14.6 | 8.4 | 0.497 | 0.066 | 6145 | 87.1 |
| | Tide 4 | 2.90 | 245 | 280 | 31.4 | 50.5 | 0.310 | 0.290 | 9654 | 24.7 |
| | Tide 5 | 1.52 | 198 | 240 | 8.2 | 7.4 | 0.466 | 0.097 | 2400 | 74.1 |
| Winter | Tide 6 | 1.77 | 214 | 288 | 8.4 | 9.3 | 0.313 | 0.094 | 2149 | 67.4 |
| | Tide 7 | 2.57 | 224 | 274 | 19.3 | 21.6 | 0.379 | 0.221 | 6625 | 42.4 |
| | Tide 8 | 1.86 | 148 | 242 | 10.4 | 12.0 | 0.454 | 0.165 | 2240 | 47.4 |

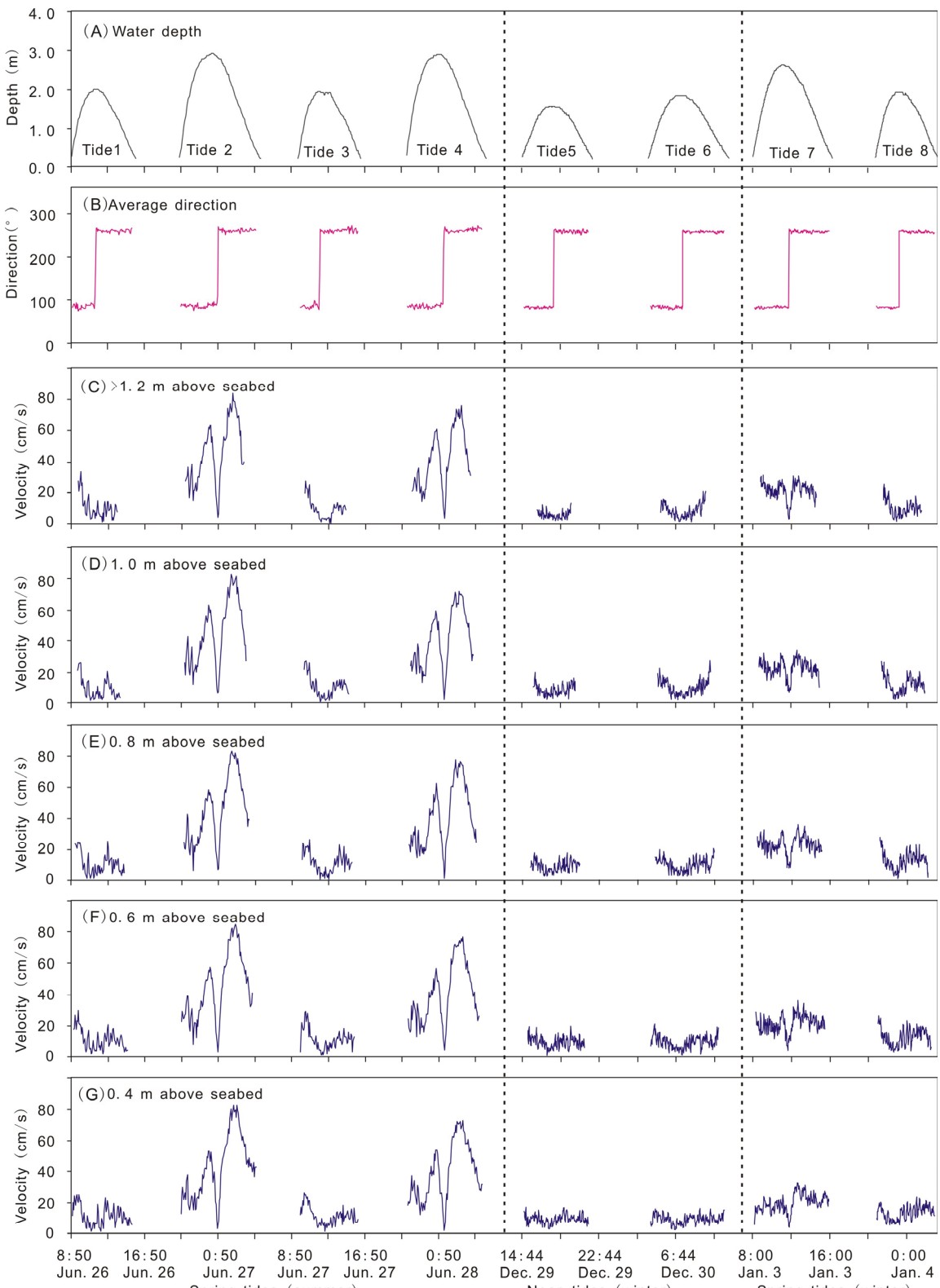

**Figure 3.** Time series of (**A**) water depth, (**B**) average flow direction, and (**C**–**G**) flow velocities at different heights above the seabed for the various seasons and tidal conditions. The velocity varied more frequently in winter than in summer because the sampling interval in winter was shorter (2 min) than that in summer (5 min). In (**C**), ">1.2 m above seabed" indicates the water surface.

Considerable differences were evident in the vertical average of the tidal current velocity over the spring and neap tides. The average vertical velocities during spring and neap tides in winter were 15.9 and 8.4 cm/s, respectively, i.e., the velocity during spring tides was $1.9\times$ greater than that during neap tides. The vertical average tidal current velocity during spring tides in summer was 26.5 cm/s, which is $1.65\times$ and $3.15\times$ greater than those during winter spring and neap tides, respectively. The tidal current velocity changed regularly during each tidal cycle, being lowest (and even close to zero) during the slack water stage, which occurred ~20 min after the high water level (Figure 3C–F). The vertical current velocity was 3–5 cm/s during the field observations and was mainly upward during the flood tide and downward during the ebb tide. The vertical current velocity tended to increase with the increase in tidal range.

*4.2. SSC Variations*

The SSC increased from the surface to the bottom (Figure 4), consistent with the typical SSC profile [36]. The SSC at the bottom was $2.2–2.9\times$ greater than that at the surface. The vertical stratification of SSC was well defined and was caused mainly by the influence of the land and vegetation on the Jiuduansha Shoal on the tidal creek and the adjacent waters; the external sea conditions had less influence. There were no extreme weather conditions during the observation period. In summer, the mean wind speed was 6.6 m/s and the maximum wind speed was 11.2 m/s from the southeast; in winter, the mean wind speed was 5.3 m/s and the maximum wind speed was 8.3 m/s from the northwest. The SSC varied regularly over a tidal cycle. Higher SSCs occurred during shallow-water stages at the beginning of the flood phase and at the end of the ebb phase (Figure 5). However, three different trends were observed within the temporal variations of SSC with changes in maximum water depth over the tidal cycle. When the maximum water depth was >2.6 m, there were two peaks of SSC at the beginning of the flood phase and the end of ebb phase (Figure 5, Tides 2 and 4). Another peak SSC value occurred when the current velocity was relatively lower during the flood slack (Figure 3). The peak in SSC was not caused by the disturbance associated with the increase in current velocity but may have been a result of the settling of suspended sediments caused by the increase in water depth (>2.0 m) and the reduced current velocity. When the maximum water depth was 2.0–2.6 m, there were only two peaks in SSC, i.e., at the beginning of the flood phase and at the end of ebb phase, and the SSC was relatively low at the high water stage (Figure 5, Tide 7). When the maximum water depth was <2.0 m there was only one peak in SSC, and this occurred at the beginning of the flood phase over the whole tidal cycle. After the start of the flood, the SSC increased rapidly, and then decreased gradually from the peak value at about 1 m water depth. The increase in SSC was not significant at the end of the ebb phase due to the weaker current velocity (Figure 5).

The maximum SSC in summer was 1.46 kg/m$^3$ and occurred at the beginning of the flood phase in the bottom layer (Figure 5). The maximum SSC in winter was 1.53 kg/m$^3$ and occurred at the end of the ebb phase, also in the bottom layer (Figure 5). The maximum tide-averaged SSCs in the surface and bottom layers were 0.129 and 0.308 kg/m$^3$, respectively, and both occurred during Tide 2 alongside the greatest water depth (2.91 m) during the observation period. The average SSC during the flood stages was higher than that during the ebb stages (Table 1). Figure 5 shows that the SSC at the early beginning of the flood phase was lower than the peak SSC at the beginning of the flood phase, which may be related to the landward movement of a water body with a low SSC that develops after the ebb phase of the previous tidal cycle.

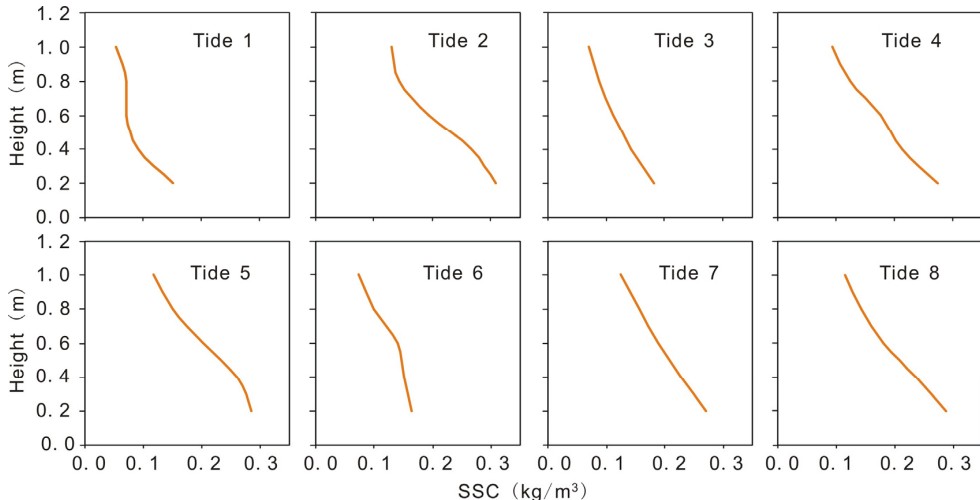

**Figure 4.** Average SSC profile for the same period of inundation.

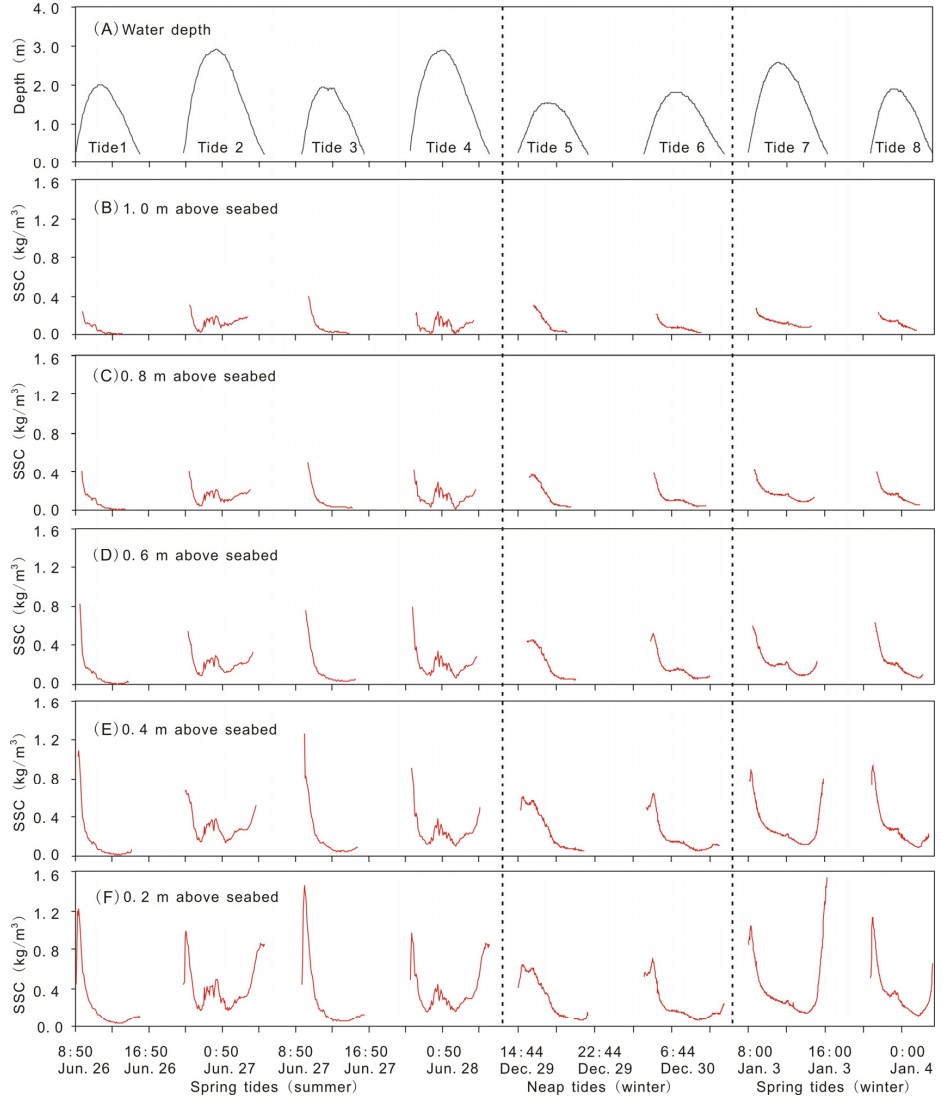

**Figure 5.** Time series of water depth (**A**), and SSC (**B–F**) at different heights above the seabed for the various seasons and tidal conditions.

### 4.3. Sediment Transport in the Tidal Creek

The discharge rate of the water transported from the tidal creek to the tidal flat varied significantly over a tidal cycle. There was an almost 10× difference in the discharge of water transported from the tidal creek during the flood and ebb phases at different tidal ranges (i.e., water depths; Figure 6B). There was a positive correlation between water discharge through the tidal creek and water depth at the observation site, which means that the greater the water depth, the greater the discharge rate. For spring tides during the summer, the discharge of water during both the flood and ebb phases exceeded 140,000 m³, with a maximum water depth of 2.91 m. For the winter neap tides, the discharge of water during the flood and ebb phases was only 7000–9000 m³, with a maximum water depth of 1.52 m (the smallest tidal range or lowest water depth of the eight tidal cycles), and the difference was more than 17×.

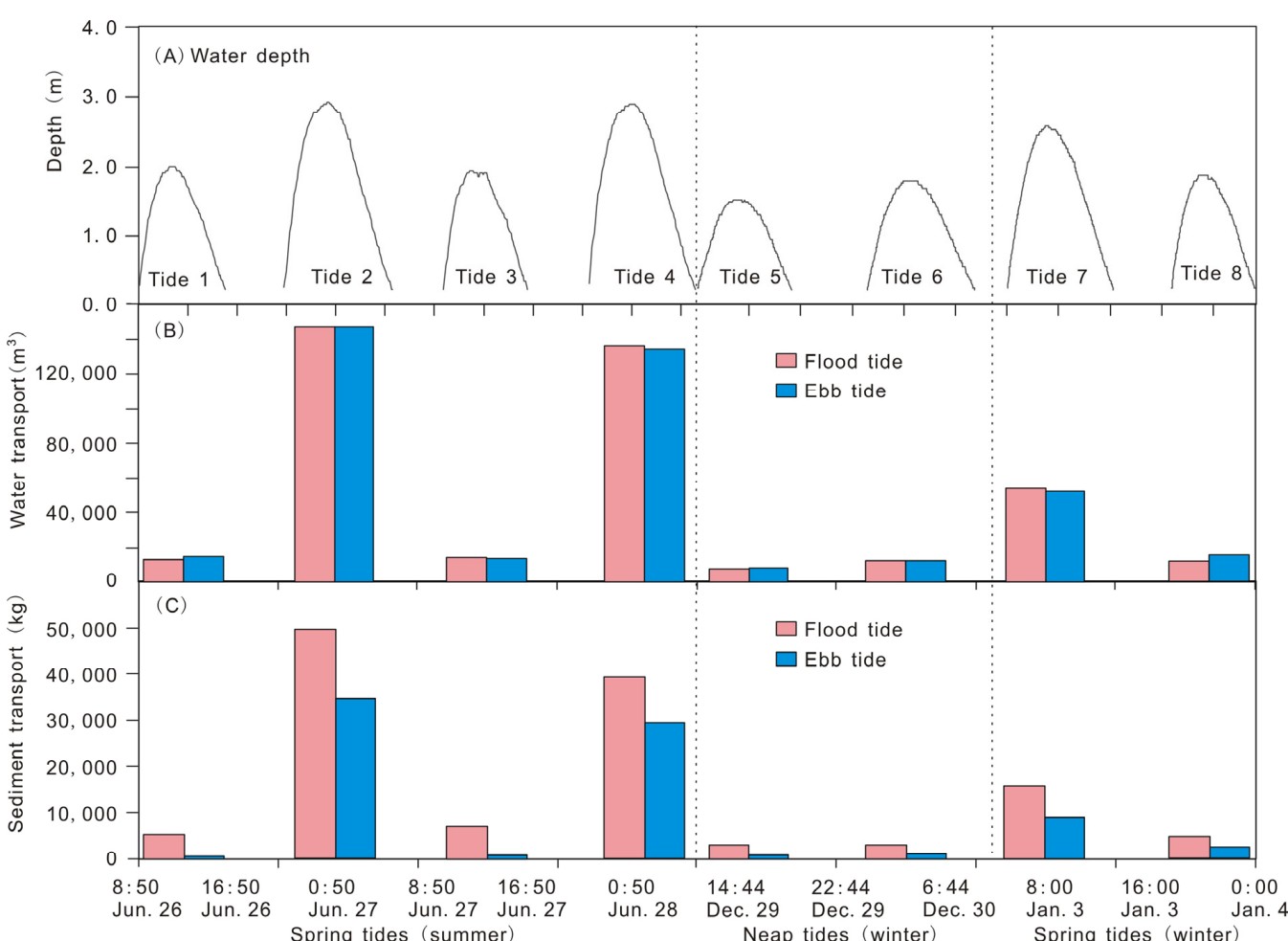

**Figure 6.** Time series of water depth (**A**), Water transport (**B**) and sediment transport (**C**) into and out of the tidal basin through the tidal creek during flood and ebb phases.

The water discharge rate through the tidal creek during the flood and ebb phases was determined from the average current velocity and the duration of the flood and ebb tides. For example, during Tide 8, the average current velocity during the flood and ebb phases was 10.4 and 12 cm/s, respectively, but the water discharge during the ebb phase was 1.28 times greater than that during the flood phase (Figure 6B), mainly because the duration of the ebb tide was significantly longer than that of the flood tide (Table 1). Although the net water discharge through the tidal creek to the tidal flat differed over each tidal cycle, the water discharge through the tidal creek in and out of the tidal flat was essentially balanced across each neap–spring cycle over the longer term.

During the tidal cycles, the net sediment discharge rates to the tidal flat through the tidal creek were all positive. The net sediment discharge transported from the tidal creek was calculated as the amount of sediment entering the tidal basin during the flood phase minus that leaving the tidal basin during the ebb phase. In general, 24.7–85.5% of the total amount of sediment discharge through the tidal creek to the tidal flat during the flood phases was trapped within the tidal basin (Figure 6C), mainly as a result of the attenuation of current velocities caused by the vegetation [24,37]. Overall, during the observation period, the greater the water depth, the greater the net sediment discharge to the tidal flat. The maximum net sediment discharge during the tidal cycles reached 15,002 kg (Figure 6C, Tide 2).

*4.4. Accretion Rate in the Tidal Basin*

The above results show that the direction of net sediment transport in the tidal creek was toward the tidal flat. Based on our observations that the net amount of sediment entering the tidal flat in winter during neap tides was 51% of that during spring tides, we also estimated the average net amount of sediment entering the tidal flat during neap tides in summer (only four spring tides were observed in summer). We assumed that the net sediment input to the tidal flat during the mean tidal period was equal to the average over the spring and neap tides, with the distribution of the spring, mean, and neap tides over a year each accounting for 1/3 of the time. Consequently, we calculated that ~37.1% of the sediment input was trapped within the tidal basin, which indicates that the tidal basin is in a phase of accretion. An old or mature tidal basin tends to enter and exit sediment balance.

The regression relation between water discharge and water depth was established using the water discharge to the tidal flat calculated from eight tidal cycles during the observation period as follows:

$$\text{WD} = 263.61 \, e^{0.0213d}, \, r = 0.99, \, n = 8, \, p < 0.001, \tag{8}$$

where WD is the water discharge entering the tidal flat through the tidal creek on a flood tide ($m^3$), d is the maximum water depth over a tidal cycle, r is the regression coefficient, *n* is the number of observations, and *p* is the level of significance. Using this relationship, together with daily tide level data from the observation station from July 2009 to June 2010, we calculated the water discharge entering the tidal flat over each tidal cycle.

The SSC in the tidal creek between July 2009 and June 2010 was estimated based on the seasonal variation of daily SSC at the observation point on the Jiuduansha Shoal [30] and the two observation periods in the tidal creek. Combined the results with the water discharge entering the tidal flat on each tidal cycle, the annual total sediment input into the tidal basin was calculated. The results showed that 9905 tons of sediment entered the tidal flat between July 2009 and June 2010, and 3675 tons (~37.1%) of the sediment remained in the tidal basin. The catchment area of the tidal basin was 0.42 $km^2$. Based on a dry bulk density of 1.3 t/$m^3$ [38] for the tidal flat sediments, indicating that the tidal basin was in an expansion phase at an accretion rate of 6.7 mm/a.

**5. Discussion**

Our results suggest that the accretion rate of the marsh on the Jiuduansha Shoal is greater than the rate of local sea level rise; therefore, the marsh is sustainable. The high accretion rate observed in this study can be attributed to the following favorable conditions. First, the Yangtze Estuary is meso–macrotidal, and sediment transport therein is active [5]. Second, the Jiuduansha Shoal is located within the turbidity maximum zone of the Yangtze Estuary. Although sediment discharge from the Yangtze River to the sea has decreased by ~70%, the SSC in the turbidity maximum zone has decreased by only ~10% because of the high background SSC and neutralization of sediment resuspension [30,39]. Third, the marsh vegetation on the Jiuduansha Shoal is dense and is, therefore, efficient in attenuating waves and currents and thus trapping sediment [24,40]. Once the sediments are deposited onto the marsh surface, they are difficult to remobilize.

We used 497 satellite images taken between 1974 and 2020 to study the development of the Jiuduansha Shoal. We found that, despite the background of a sharp decrease in sediment entering the estuary from the Yangtze River, the Jiuduansha Shoal continued to accrete owing to the coarsening fluvial sediment, the sediment supply from outside the estuary, the approximate cone shape of the Jiuduansha Shoal, and the siltation-promoting effects of vegetation [29]. From 1990 to 2020, the growth rate of the low tide area was 3.0 km$^2$/a; however, the growth rate decreased from 2010 to 2020 [29]. After the construction of the southern training jetty of the Deep-Water Channel Project, the hydrodynamic forces on the northern side of the Jiuduansha Shoal were weakened and the exchange of water and sediment with the North Passage was hindered, and deposition on the northern side of the Jiuduansha Shoal accelerated [41,42]. The growth of the low tide area of the Jiuduansha Shoal was also adversely affected [29]. However, with the decrease in sediment discharge from the Yangtze River, there was a degree of erosional retreat on the southern Jiuduansha Shoal and the southeast of tail of Jiuduansha Shoal [42], and substantial erosion also occurred on the southern side of the Upper Shoal [43].

Around 50% of the world's deltas have a mesotidal to macrotidal regime, similar to that of the Yangtze Delta [5]. Therefore, tidal currents may erode subaqueous delta and mudflat areas, and the resuspended sediments may be transported into marshes in these types of delta systems, which are threatened by fluvial sediment decline and sea level rise. However, tidal marshes do not always persist under a regime of sea level rise and fluvial sediment decline. For example, ~7000 km$^2$ (or 25% of the total) of the marsh wetlands on the Mississippi Delta has sunk below sea level over recent decades, and an additional 10,000–13,500 km$^2$ is expected to be lost by the end of this century [44]. In contrast to the Yangtze Delta, the Mississippi Delta is microtidal (i.e., tidal range < 0.5 m) and has lower SSCs. As a result, sediment transport into the Mississippi marshes is insufficient to sustain a deposition rate that is equivalent to (or greater than) the rate of sea level rise [5]. Estuarine management under accelerating sea level rise is likely to become increasingly complex [45]. Thus, the resistance of a tidal marsh to the combined impacts of rising sea levels and falling fluvial sediment inputs depends on environmental conditions such as the tidal regime, sediment supply, and sediment properties, which determine the ability of the marsh to sustain accretion.

In conclusion, it can be inferred that the middle–high salt marsh on the Jiuduansha Shoal will continue to accrete, but the rate of this accretion may decrease, and the salt marsh silting up of the entire sand body in the plane direction would be slowed down or even eroded. As a national nature reserve, the overall stability of the sand body must be ensured. Although the development of the Jiuduansha Shoal itself is less affected by human activities, it is still heavily influenced by the exploitation of the surrounding waters. To ensure the stability of Jiuduansha Shoal sand body, the comprehensive study of natural and human activities will be further strengthened. It will be necessary to expand observations of, and research into, the development and evolution of the Jiuduansha Shoal to provide theoretical support for the development of protection and management strategies.

Due to the cost of observation, only one tidal channel system in Jiuduansha Shoal had been observed, and only 8 tidal cycles have been observed. The tidal creeks system is complex in the Jiuduansha Shoal. At the same time, because of its large area, the development of the tidal marsh has remarkable spatial characteristics. Thus, using the change of sedimentation in a tidal basin to infer the whole Jiuduansha Shoal, which exist errors inevitably. However, the study and observation of the developmental trend of Jiuduansha Shoal still have a strong reference value. In the future, we will carry out systematic observation and research on Jiuduansha Shoal with new observation methods.

## 6. Conclusions

In a tidal creek on the Jiuduansha Shoal, the flow velocity varied between 0 and 84 cm/s (average 19.5 cm/s), and the SSC ranged from 0.034 to 1.53 kg/m$^3$ (average 0.279 kg/m$^3$). The SSC during the flood phase was markedly higher than that during the ebb phase, leading to a sediment retention rate of 37.1%. The marsh accretion rate was estimated to be 0.0092 mm/tide and 6.7 mm/a, which exceeds the relative rate of sea level rise in the Yangtze Delta (~4 mm/a). Thus, sediment supply from the turbid zone of the Yangtze Estuary will be able to sustain the marsh under rising sea levels and falling fluvial sediment inputs. This study helps to improve our understanding of how tidal marshes will respond to rising sea levels and declining fluvial sediment supply.

**Author Contributions:** Formal analysis, B.S.; methodology, L.L.; software, S.W.; validation L.K.; formal analysis, G.W.; investigation, W.Z. and J.H.; writing—original draft preparation, P.L.; writing—review and editing, W.Z. All authors have read and agreed to the published version of the manuscript.

**Funding:** This study was funded by the Open Research Fund of State Key Laboratory of Estuarine and Coastal Research (SKLEC-KF202004), Open Research Fund of Key Laboratory of Coastal Science and Integrated Management, Ministry of Natural Resources (2021COSIMZ001), Laizhou-Laixi Concentration Area Ecological Restoration Support Investigation Project (ZD20220220) and the Research Funds of Happiness Flower ECNU (20212110).

**Institutional Review Board Statement:** Not applicable.

**Informed Consent Statement:** Not applicable.

**Data Availability Statement:** Not applicable.

**Acknowledgments:** The authors would like to thank Ang Gao for Field observation and two anonymous reviewers for their detailed comments that helped improve the paper greatly.

**Conflicts of Interest:** The authors declare no conflict of interest.

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
