# Peer review of "Tidal Sediment Supply Maintains Marsh Accretion on the Yangtze Delta despite Rising Sea Levels and Falling Fluvial Sediment Input"

_water, doi:10.3390/w14223768_

Round 1

Reviewer 1 Report

I enjoyed reading the manuscript and think it is publishable if the following issues are addressed. 

Please better explain the basis for determining the amount of sediment retained. I think the authors are concluding that sediment on outgoing tide minus sediment on incoming tide = sediment retained, but this isn't entirely clear. The description of the field measurements and the methodology in general are good, but this point is lacking. Also a better understanding of the error or confidence intervals for the estimates of sediment in and out would be helpful. A table with a column showing what they believe to be the net deposition per tidal cycle along with error estimates would strengthen this paper considerably (e.g., add to table 1).

While I ageee with the authors that vegetation helps retain sediment, which is a main conclusion of the paper, I did not see much discussion or citations in the introduction supporting such a statement, and their data didn't really demonstrate that very well because almost the entirety of the estuary was covered in vegetation, so they were unable to compare sediment accumulation rates in the absence of vegetation. I think it would greatly strengthen the paper if they looked to the literature to better describe how and why vegetation retains sediment and also look to the literature to find examples of aggradation rates in the absence of vegetation. The authors operate on what I think is a reasonable premise, that marshes with vegetation retains sediment at a higher rate than "controls" without vegetation, but they provide no data nor cite much literature to support that finding or assertion. I suggest they look to the literature in both freshwater and saltwater environments to provide better support for this conclusion.

Author Response

Reviewer: 1

Comments and Suggestions for Authors

I enjoyed reading the manuscript and think it is publishable if the following issues are addressed.

Please better explain the basis for determining the amount of sediment retained. I think the authors are concluding that sediment on outgoing tide minus sediment on incoming tide = sediment retained, but this isn't entirely clear. The description of the field measurements and the methodology in general are good, but this point is lacking. Also a better understanding of the error or confidence intervals for the estimates of sediment in and out would be helpful. A table with a column showing what they believe to be the net deposition per tidal cycle along with error estimates would strengthen this paper considerably (e.g., add to table 1).

 [Authors’s reply: Thanks for this suggestion! In this paper, the calculation method of the sediment discharge detained in tidal flat is an ideal and the simplest method. For example, at the beginning of flood tide and at the end of ebb tide phase, the data can not be monitored, and whether there is water exchange with other tidal channel systems at high water levels, but this amount is relatively small compared to the amount of sediment entering and leaving the tidal flat during the whole tidal cycle, and this has been explained in section 3.2. This paper focused on the developmental and evolutionary trends of tidal flat. Although there are errors in the calculation of the discharge sediment of each tidal cycle during the observation period, the effects of these errors will be filtered out in one-year time, therefore, the discharge sediment of each tidal cycle is not estimated in this paper. See section 3.2. We add two columns in Table 1, Net deposition (kg) and Net Sediment discharge rates (%).]

While I agree with the authors that vegetation helps retain sediment, which is a main conclusion of the paper, I did not see much discussion or citations in the introduction supporting such a statement, and their data didn't really demonstrate that very well because almost the entirety of the estuary was covered in vegetation, so they were unable to compare sediment accumulation rates in the absence of vegetation. I think it would greatly strengthen the paper if they looked to the literature to better describe how and why vegetation retains sediment and also look to the literature to find examples of aggradation rates in the absence of vegetation. The authors operate on what I think is a reasonable premise, that marshes with vegetation retains sediment at a higher rate than "controls" without vegetation, but they provide no data nor cite much literature to support that finding or assertion. I suggest they look to the literature in both freshwater and saltwater environments to provide better support for this conclusion.

[Authors’s reply: Thanks for this suggestion! Vegetation plays an important role in slowing down hydrodynamic increase of sediment deposition. Li et al [24] studied the retention of suspended sediment by different plant species (S. Mariqueter, S. alterniflora, and P. australis) in the Yangtze estuary. The results show that the vegetation can absorb part of the suspended fine sediment. At the same time, due to the hydrodynamic attenuation caused by the vegetation, the suspended sediment in the water body increases. Thus, the suspended sediment caused by the vegetation adsorption and hydrodynamic attenuation plays a very important role in the marsh sedimentation. The Spartina alterniflora were planted in 1997, which promoted the silting up of the Middle and Lower Shoal [27, 28], and the silting rate of Jiuduansha Shoal obviously increased [23, 29]. This paper referred to this work in Section 1 and Section 5. Because this paper focused on the comprehensive deposition rate of the tidal marshes, not much attention to the vegetation on the acceleration of sediment deposition, and only explained qualitatively. In the future, we will strengthen the observation and research in this area to better reveal the mechanism of tidal marshes development.]

Reviewer 2 Report

The manuscript by Li et al. investigates the survivability of tidal marshes in the Yangtze Delta under the combined influence of rising mean sea level and decreasing sediment supply. The results presented are potentially of interest, in my opinion, after major revisions. The manuscript, though well written, can be improved by being more rigorous in its analysis. My comments are detailed below.

1. L46: The sentence reads as if survival of marshes only depends on the sediment transport, whereas in reality it depends on several factors such as presence/absence of physical barriers, accommodation/migration space, etc.  

2. The Introduction needs to be re-focused to set up the questions and objectives of the paper. Although the research questions are presented, it is unclear how they are built upon previous research. This will help better clarify the novelty of this work.

3. L145-146: Did you mean T1 and T2?

4: L218: Change to “spring tides”.

5. Section 4: It is unclear how authors justified that only 8 tidal cycles are enough to represent the hydrodynamics of the system including current speeds, sediment dynamics, etc. This is an important aspect which is currently missing and should be explained in detail.

6. Section 5: There is an ongoing debate around tidal marsh survivability and upper end sea level rise scenarios. While current sea level rise considered is 4 mm/year, the potential RCP8.5 scenario suggests a rate of nearly 15mm/year. I’d suggest authors expanding on the discussion on potential future scenarios where the tidal marshes can either survive or degrade depending on the pathway taken. Further, survivability of tidal marshes depends on human disturbances and management schemes too which were not considered here. See following references as suggestions to enrich the discussion:

a) Thresholds of mangrove survival under rapid sea level rise

b) Sea level rise impacts on estuarine dynamics: A review

7. General comment: The paper should be up-front and clearly state the limitations associated with this study and direct future efforts.

Author Response

Reviewer: 2

Comments and Suggestions for Authors

The manuscript by Li et al. investigates the survivability of tidal marshes in the Yangtze Delta under the combined influence of rising mean sea level and decreasing sediment supply. The results presented are potentially of interest, in my opinion, after major revisions. The manuscript, though well written, can be improved by being more rigorous in its analysis. My comments are detailed below.

  1. L46: The sentence reads as if survival of marshes only depends on the sediment transport, whereas in reality it depends on several factors such as presence/absence of physical barriers, accommodation/migration space, etc.

[Authors’ reply: Thanks for this suggestion! The sediment transport is only one major factor for the survival of marshes. We have done as suggested. See line 46.]

  1. The Introduction needs to be re-focused to set up the questions and objectives of the paper. Although the research questions are presented, it is unclear how they are built upon previous research. This will help better clarify the novelty of this work.

[Authors’ reply: Thanks for this suggestion! We have reorganized the questions in the introduction and made the topic of the article clearer. See lines 48-52, lines 59-61 and lines 78-83.]

  1. L145-146: Did you mean T1 and T2?

[Authors’ reply: Revised as suggested. See line 169.]

4: L218: Change to “spring tides”.

[Authors’ reply: Revised as suggested. See line 245.]

  1. Section 4: It is unclear how authors justified that only 8 tidal cycles are enough to represent the hydrodynamics of the system including current speeds, sediment dynamics, etc. This is an important aspect which is currently missing and should be explained in detail.

[Authors’ reply: Thanks for this suggestion! Due to the limitation of observation funds, only 8 tidal cycles were observed, including spring and neap tides in winter and summer. In general, the observation results can represent the hydrodynamic characteristics and siltation characteristics of the tidal flat in winter and summer. The spring and fall characteristics were between winter and summer, and we have mentioned in Section 4.4. In this paper, we focused on the developmental and evolutionary trend of tidal marshes. In the absence of long-term and continuous observation data, the influence of tidal cycle and seasonal errors would be filtered out in the calculation of one-year time. Therefore, the annual sedimentation rate estimated from these 8 tidal cycles is relatively scientific and credible. We have done as suggested in Section 3.1. See lines 152-159.]

  1. Section 5: There is an ongoing debate around tidal marsh survivability and upper end sea level rise scenarios. While current sea level rise considered is 4 mm/year, the potential RCP8.5 scenario suggests a rate of nearly 15mm/year. I’d suggest authors expanding on the discussion on potential future scenarios where the tidal marshes can either survive or degrade depending on the pathway taken. Further, survivability of tidal marshes depends on human disturbances and management schemes too which were not considered here. See following references as suggestions to enrich the discussion:
  2. a) Thresholds of mangrove survival under rapid sea level rise
  3. b) Sea level rise impacts on estuarine dynamics: A review

[Authors’ reply: Thanks for this suggestion! The two papers are very good, mainly to study the impact of sea level rise on the estuarine dynamic system. The Jiuduansha Shoal is a national nature reserve. Although the development of the Jiuduansha Shoal itself is less affected by human activities, it is still heavily influenced by the exploitation of the surrounding waters. To ensure the stability of Jiuduansha Shoal sand body, the comprehensive study of natural and human activities will be further strengthened. See section 5, lines 410-411 and lines 419-422. ]

  1. General comment: The paper should be up-front and clearly state the limitations associated with this study and direct future efforts.

[Authors’ reply: Thanks for this suggestion! Because of the cost of observation, only one tidal channel system in Jiuduansha Shoal had been observed, and only 8 tidal cycles have been observed. The tidal creeks system is complex in the Jiuduansha Shoal. At the same time, because of its large area, the development of the tidal marsh has remarkable spatial characteristics. Thus, using the change of sedimentation in a tidal basin to infer the whole Jiuduansha Shoal, which exist errors inevitably. However, the study and observation of the developmental trend of Jiuduansha Shoal still have a strong reference value. In the future, we will carry out systematic observation and research on Jiuduansha Shoal with new observation methods. See section 5, lines 426-433.]

Round 2

Reviewer 2 Report

The authors did a good job of addressing the comments raised in an earlier version of this manuscript and therefore, I am happy to recommend this research for publication. Well done!